# Development of Permanently Installed Magnetic Eddy Current Sensor for Corrosion Monitoring of Ferromagnetic Pipelines

**Rukhshinda Wasif [1,\*], Mohammad Osman Tokhi [1], Gholamhossein Shirkoohi [1,2], Ryan Marks [2] and John Rudlin [2]**

1   School of Engineering, London South Bank University, London SE1 0AA, UK; tokhim@lsbu.ac.uk (M.O.T.); maziar.shirkoohi@lsbu.ac.uk (G.S.)
2   The Welding Institute Limited, Cambridge CB21 6AL, UK; ryan.marks@twi.co.uk (R.M.); john.rudlin@hotmail.co.uk (J.R.)
\*   Correspondence: wasifr@lsbu.ac.uk

**Abstract:** Permanently installed sensors are a cost-effective solution for corrosion monitoring due to their advantages, such as less human interference and continuous data acquisition. Some of the most widely used permanently installed corrosion sensors are ultrasonic thickness (UT) gauges. However, UT sensors are limited by the need for coupling agents between pipe surfaces and sensors. The magnetic eddy current (MEC) method, on the other hand, does not require couplant and can be used over insulations. With the development of powerful rare earth magnets, MEC sensors with low power consumption are possible, and there is the prospect of using them as permanently installed sensors. A novel wireless magnetic eddy current sensor has been designed and optimized using finite element simulation. Sensitivity studies of the sensors reveal that the excitation frequency is a critical parameter for the detection of corrosion defects. An in-depth explanation of the relationship between the sensitivity of the sensor and the excitation frequency is presented in this paper. The results of an accelerated corrosion test, conducted to simulate the service environment of the sensor, are also discussed. It was observed that the sensor signals are very sensitive to corrosion defects and show no subtle differences due to temperature and humidity changes.

**Keywords:** corrosion monitoring; magnetic eddy current; permanently installed sensor; wireless sensor; oil and gas pipelines; magnetic flux



## 1. Introduction

Corrosion is one of the leading causes of catastrophic failures in mild carbon steel structures. According to the NACE report, corrosion costs almost 3–4% of the GDP of each nation. The global cost of corrosion is estimated to be £2.5 trillion US dollars [1]. EGIG reported internal corrosion as the second major cause of incidents in gas pipelines for the period 1970–2019 [2]. The US Department of transportation, pipeline, and hazardous safety material report indicated a three-fold increase in the internal corrosion incidents in oil pipelines during the period of 1970 to 2013. Similar trends were observed for gas pipelines [3]. The above-mentioned facts and figures show that corrosion is a critical issue and becomes severe with the aging of the pipelines. Continuous corrosion monitoring is, therefore, substantial to avoid economic, environmental, and human losses.

A number of non-intrusive techniques are employed in the industry for monitoring corrosion. In-line inspection tools such as pipeline inspection gauges (PIGs) or robotic crawlers are commonly used for periodic inspection of corrosion. They use non-destructive evaluation (NDE) techniques such as ultrasonic, electromagnetic acoustic transducers (EMATs), eddy current, magnetic flux leakage (MFL), pulsed eddy current, or acoustic emission (AE) for detecting various types of defects in pipelines. The in-line inspection tools are expensive and require special pipe configurations [4]. They can only be deployed

in long pipes with a certain pressure and no sharp bends, valves, or branches [5–7]. Despite the extensive inspections, in-line tools are, therefore, run periodically after 5–7 years [8]. On-line inspection techniques, such as manual UT tools, are reliable and have high resolution for corrosion defects [9]. Online inspection requires access to the pipes, and there are high costs associated with scaffolding and rope access [10]. Additionally, the manual UT inspection is influenced by surface roughness, and the cost of surface preparation is a major drawback. [11].

In recent years, permanently installed wireless sensors have become popular in the industry because of their advantages such as low cost, continuous data acquisition, less human interference, and reliable data since the transducers are fixed at the inspection points [12]. A number of wireless corrosion sensors have been developed and discussed in the literature. Radiofrequency identifier (RFID) sensors and interdigitated capacitance-based sensors are based on the relative movement of two components of the sensors [13]. They need access to both sides of the pipe and have an indirect mechanism for corrosion detection [14–16]. Surface acoustic wave (SAW) sensors and guided wave permanently installed sensors are sensitive to corrosion defects, but they are highly affected by environmental parameters such as temperature [17,18]. Ultrasonic thickness gauges are commonly used permanently installed sensors [19]. They can detect buried corrosion defects, and the confidence level of UT tools is around 95% [11]. The major disadvantage of the UT sensors is that they require coupling agents to be installed between the pipe surface and the transducers. The magnetic methods, on the other hand, do not require coupling agents. Magnetic sensors, for instance, anisotropic magneto-resistive (AMR) sensors to detect corrosion by measuring the deflection of the low-frequency alternating current (AC) injected into the pipeline at different locations, have been reported [20]. Techniques such as alternating current field method and frequency-dependent magnetic field induced due to AC current injected at remote contact points are also discussed [21]. These approaches require the pipes to be injected at different locations and also may have limited sensitivity. Magnetic NDE techniques such as MFL and MEC are considered reliable for identifying corrosion defects in ferromagnetic pipelines. With the development of powerful rare earth magnets, there is the possibility of using the MFL or MEC sensors as permanently installed sensors due to considerably reduced probe size and low power requirements. The magnetic eddy current method, which is an adaptation of magnetic flux leakage and eddy current technique, requires low magnetization levels and has good sensitivity for detecting internal corrosion in ferromagnetic structures [22,23]. Additionally, the technique has the capability to distinguish between internal and external corrosion, which is not possible with the traditional MFL [24]. Moreover, the MFL method relies on the detection of the leakage magnetic field using Hall effect sensors, which are manufactured from semiconductors. The performance of semiconductors is highly affected by temperature variations, whereas coil sensors in MEC, being passive in nature, are more tolerant to changes in temperature.

MEC sensors have been found to be promising for the detection of narrow-surface and buried defects such as cracks by measuring the perturbations in the permeability of the test sample [25–27]. Most of the research on MEC sensors has focused on the effect of the magnetization level and defect quantification using different signal parameters [22–24]. A wide range of detection frequencies are used for MEC testing. Apart from the magnetization level in the test sample, the sensitivity of the sensor is predominantly affected by the AC characteristics such as excitation frequency and voltage [28]. A wide range of inspection frequencies (1 kHz–500 kHz) are used in the above-mentioned studies, but the reasons for selecting a particular frequency are not explained. The effectiveness of the MEC sensors is well established in the literature, but the detection mechanism and the frequency-related sensitivity of the technique are still unclear.

The research work presented in this paper has two novelty aspects: (1) the study on the feasibility of developing a low-power wireless MEC sensor for permanent installation to monitor internal corrosion and (2) a comprehensive description of the relationship between the sensitivity of the sensor and the AC excitation frequency.

The rest of the paper is arranged as follows: Section 2 covers the working principle of the MEC sensor, Section 3 presents the studies on sensitivity optimization using FEA and experiments, Section 4 illustrates the design of the wireless sensor and results of the accelerated corrosion test, Section 5 includes the discussions, Section 6 concludes the research work, and Section 6 suggests future research.

## 2. Working Principle

The basic principle of the magnetic eddy current technique is based on the nonlinear relationship between the magnetization field ($H$) and the induced magnetic flux density ($B$) in ferromagnetic materials such as mild carbon steel 1002, as shown in Figure 1. When a direct current (DC) magnetization field is imposed on a mild steel sample, the increase in the induced magnetic field is higher than the applied field ($H$) due to the presence of unpaired electrons. Initially, there is an increase in the magnetic permeability ($\mu$). However, beyond a certain point (a) in the curve, the magnetic induction does not increase at the same level as $H$ because of the alignment of the unpaired electrons in the direction of the applied magnetization field. Consequently, there is a distinct decline in permeability. The relationship between $B$ and $H$ is given as

$$B = \mu_0(H + x_m H) = \mu_0 H(1 + x_m) = \mu H \tag{1}$$

where $x_m$ is the magnetic susceptibility and $\mu$ is the magnetic permeability of the ferromagnetic material.

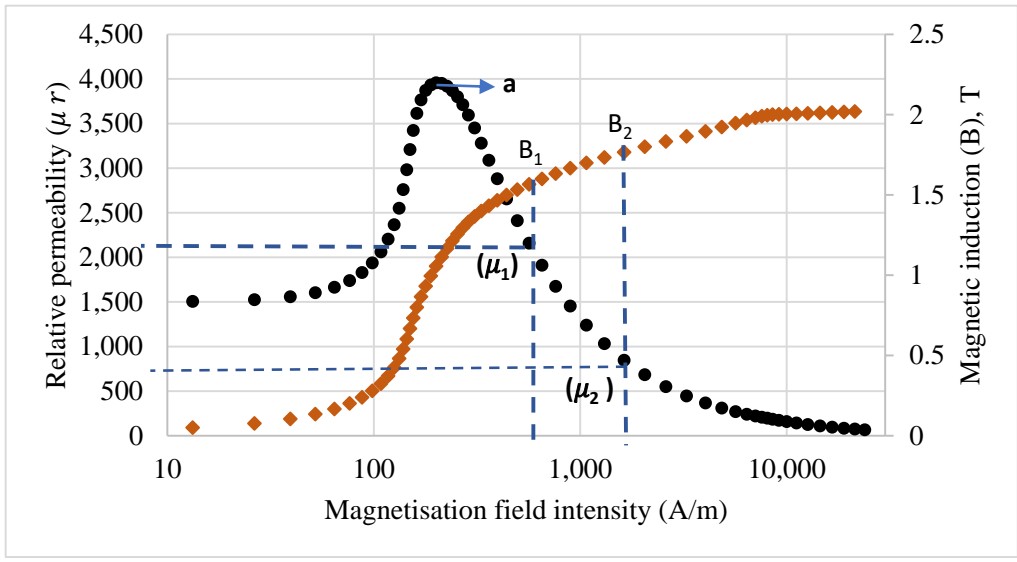

**Figure 1.** Magnetization and relative permeability as a function of applied field.

When a steel sample is magnetized beyond the point (a) in the Figure 1, there is a leakage magnetic flux density $B_{lmf}$, whereas the magnetic flux density inside the uncorroded sample is $B_1$. If there is a decrease in the wall thickness of the sample owing to corrosion, there will be an increase in the $B_{lmf}$ and the flux density inside the sample will rise from $B_1$ to $B_2$ correspondingly decreasing the magnetic permeability from $\mu_1$ to $\mu_2$.

The mechanism of the magnetic eddy current principle for the detection of corrosion can be explained by the lumped parameter reluctance network. Magnetic circuits may be viewed as simplified electric circuits, in which each component provides reluctance to the flow of magnetic flux, similar to electrical resistance in an electric circuit. However, contrary to electric circuits, the magnetic flux is not contained within the magnetic path. As the flux spreads in the air gap that surrounds the circuit components, it is referred to as the fringe field. The simplified magnetic circuit model is shown in Figure 2.

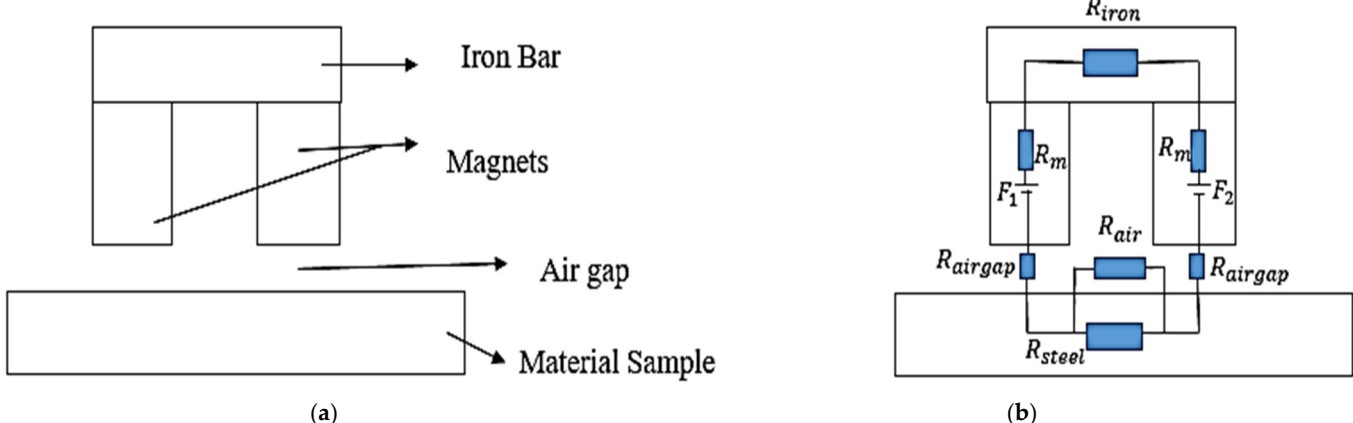

**(a)**                                          **(b)**

**Figure 2.** (**a**) The schematic representation of the magnetization yoke, and (**b**) the simplified equivalent magnetic circuit using lumped parameters reluctance network.

$F_1$ and $F_2$ represent the magneto motive force applied by the magnets which serve as the source of magnetic flux. The magnetic flux in any component of the circuit can be calculated using

$$\varnothing = \frac{F}{R} \tag{2}$$

where $R$ is the reluctance and is computed from Equation (3).

$$R = \frac{l_l}{\mu A} \tag{3}$$

where $l_l$ is the length of the flux path, $\mu$ is the magnetic permeability, and $A$ is the cross-sectional area through which flux flows. The corrosion of the sample under test results in a decrease in the cross-sectional area, subsequently increasing the reluctance of the sample. As a result, the magnetic permeability of the sample is decreased. The change in permeability can be detected by monitoring the perturbations in the impedance of the harmonically excited coil placed above the magnetized sample.

According to Kirchhoff's law, the analytical expressions for computing the flux in the sample and leakage magnetic flux using the equivalent magnetic circuit can be written as:

$$\varnothing_{plate} = \frac{F_1 + F_2}{2R_m + R_{iron} + 2R_{airgap} + R_{plate}} \tag{4}$$

$$\varnothing_{lmf} = \frac{F_1 + F_2}{2R_m + R_{iron} + 2R_{airgap} + R_{plate} + R_{air}} \tag{5}$$

Considering that the reluctance of air is much larger than that of a mild steel plate, the leakage magnetic flux can be ignored when calculating coil impedance because it is much smaller.

When a coil with sinusoidal excitation is placed above the magnetized steel sample, the magnetization field will affect the eddy current coil signal in two ways. Firstly, the depth of penetration ($\Delta$) of the eddy currents is increased by a few millimeters, as expressed by

$$\Delta = \sqrt{\frac{1}{\pi f \mu \sigma}} \tag{6}$$

where, $f$ is the excitation frequency, $\sigma$ is the electrical conductivity, and $\mu$ is the magnetic permeability of the sample. The depth of penetration of eddy currents is limited to a fraction of a millimeter of mild steel owing to its high permeability. However, magnetizing the sample can increase it by a few millimeters due to the reduced permeability.

Secondly, the eddy currents field, set up in the sample due to the time-harmonic magnetic field of the coil, will be affected by the perturbations in the permeability caused by the corrosion defects. Consequently, a variation in the impedance of the coil will be experienced due to the corrosion of the sample.

The impedance of the coil ($Z = R + X_L$), which is the resistance to the flow of the alternating current in the coil, is composed of two components. The resistance ($R$) of the coil is a function of the electric field set up by the eddy currents and is sensitive to the variation in the conductivity ($\sigma$) of the sample. The imaginary part of the impedance, called reactance ($X_L$), is the opposition due to magnetic fields induced by the time-varying currents and is affected by the permeability changes of the test sample. Theoretically, the expression for calculating the perturbation in the coil impedance ($\Delta Z$) placed above the samples with different metallic composition is given as follows [29,30]:

$$\Delta Z = Kj\omega \int_0^\infty \left( \frac{P2\,(r_{2,}\,r_{1)}}{\alpha^6} \right) A(\alpha) \left[ \varnothing(\alpha) - \frac{\alpha\mu_2 - \alpha\mu_0}{\alpha\mu_2 + \alpha\mu_0} \right] \tag{7}$$

where $K$ and $P$ are the constants related to the coil parameters; and $A(\alpha)$, and $\varnothing(\alpha)$ are the magnetic scalar potential and flux density set up by the coil, respectively, and are a function of $\alpha$, which is related to both conductivity and permeability of the test sample.

$$\alpha_i = \sqrt{\alpha + (j\omega\mu_i\sigma_i)} \tag{8}$$

Under DC magnetization, the conductivity effect is found to be negligible and the real part of the impedance can, therefore, be ignored for the sensor optimization study [22,26].

## 3. Sensitivity Optimization

### 3.1. Finite Element Modelling

In this section, the optimization of the magnetic eddy current sensor was carried out by simulations on Comsol Multiphysics. The problem of imposing an alternating current field on a DC bias magnetic field is governed by two different sets of equations. A two-step approach of the magnetic transient study was adopted to optimize the sensitivity of the sensor while achieving a smaller probe size.

#### 3.1.1. Magneto-Static Modelling

The magnetization level in the sample is a significant parameter to achieve higher sensitivity. While under magnetization may lead to a missed defect signal, oversaturation not only increases the size of the sensor but also the background noise. A 3-D magnetization yoke was modeled in the magnetic fields with no currents module to optimize the size parameters without compromising the sensitivity. The magnetization fields of the permanent magnets in a medium are computed by evaluating the magnetic scalar potential ($\nabla Vm$).

$$H = -\nabla Vm \tag{9}$$

The constitutive relation for solving the problem in a particular medium is given by Equation (1), i.e., the B-H curve of the mild steel. The BH curve of Carbon Steel 1002 from the material library was used to calculate the magnetic flux density for points equidistant located at 1 mm in the test sample. The material properties defined for different components in the model are listed in Table 1.

A uniform corrosion of 1 mm up to 4 mm with decrement steps of 1 mm was simulated in the model, and their relative magnetic permeability values were evaluated from the BH curve.

**Table 1.** The material properties defined in the FE model.

| Description | Material | Properties | |
|---|---|---|---|
| Magnets | Neodymium | Magnetization: | 835 kA/m |
| | | Relative Permeability: | 1.05 |
| | | Conductivity | $1 \times 10^6$ S/m |
| Iron Bar | Cast Iron | Relative Permeability: | 4000 |
| | | Electrical Conductivity | $11.2 \times 10^6$ S/m |
| Plate Sample | Low-carbon steel 1002 | Relative Permeability: | BH curve |
| | | Electrical Conductivity | $8.41 \times 10^6$ S/m |

A mesh convergence study was performed by comparing flux at a fixed point for different mesh sizes. An adaptive meshing technique was used, and mesh parameters were refined until no difference in flux was observed after further mesh refinement. Figure 3 shows the point at which the flux was calculated to carry out the mesh convergence, and the graph between the maximum element size and error between the successive flux computed for the respective mesh sizes. The mesh parameters defined in the model are listed in Table 2.

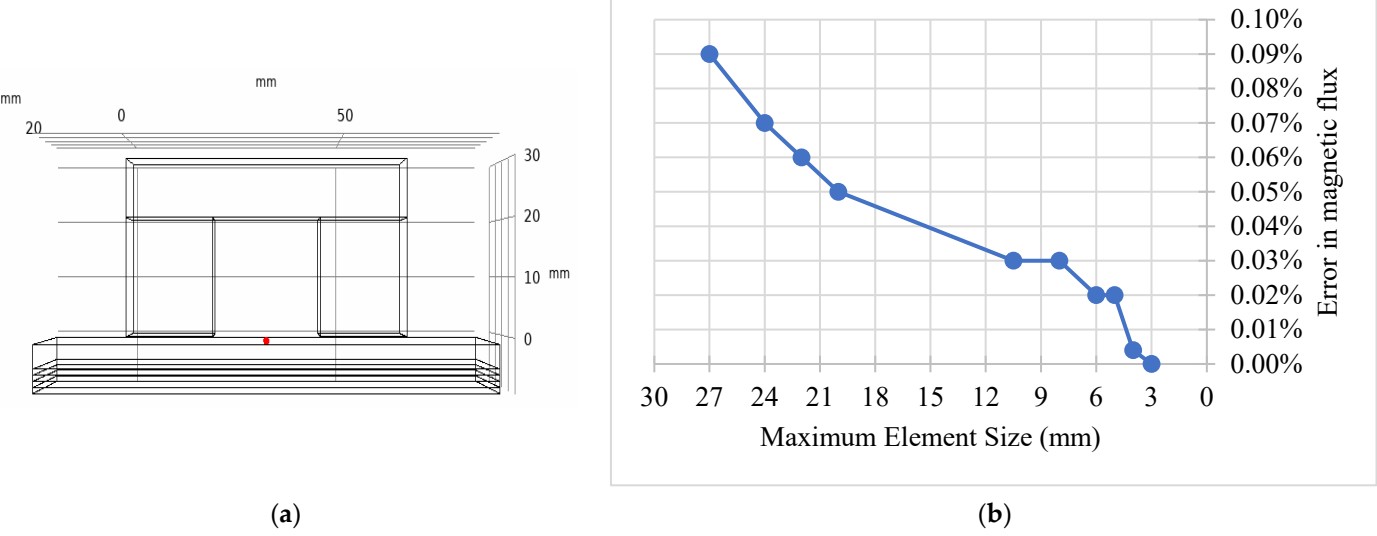

(**a**)

(**b**)

**Figure 3.** (**a**)The location of the point at which the magnetic flux was computed for mesh convergence study, and (**b**) the graph showing the decrease in the error between the flux values with mesh refinement.

**Table 2.** The mesh parameters obtained from mesh convergence study.

| Element Size Parameters | Values |
|---|---|
| Maximum Element Size | 3 |
| Minimum Element Size | 0.05 |
| Maximum Element growth rate | 1.3 |
| Resolution of narrow regions | 1 |

It was found that a minimum flux density of 1.4 T in the sample can give good sensitivity using a probe size of $40 \times 65 \times 40$ mm. The dimensions of the probe are illustrated in Figure 4a. The FE model along with the computed flux densities and relative permeability are shown in Figure 4.

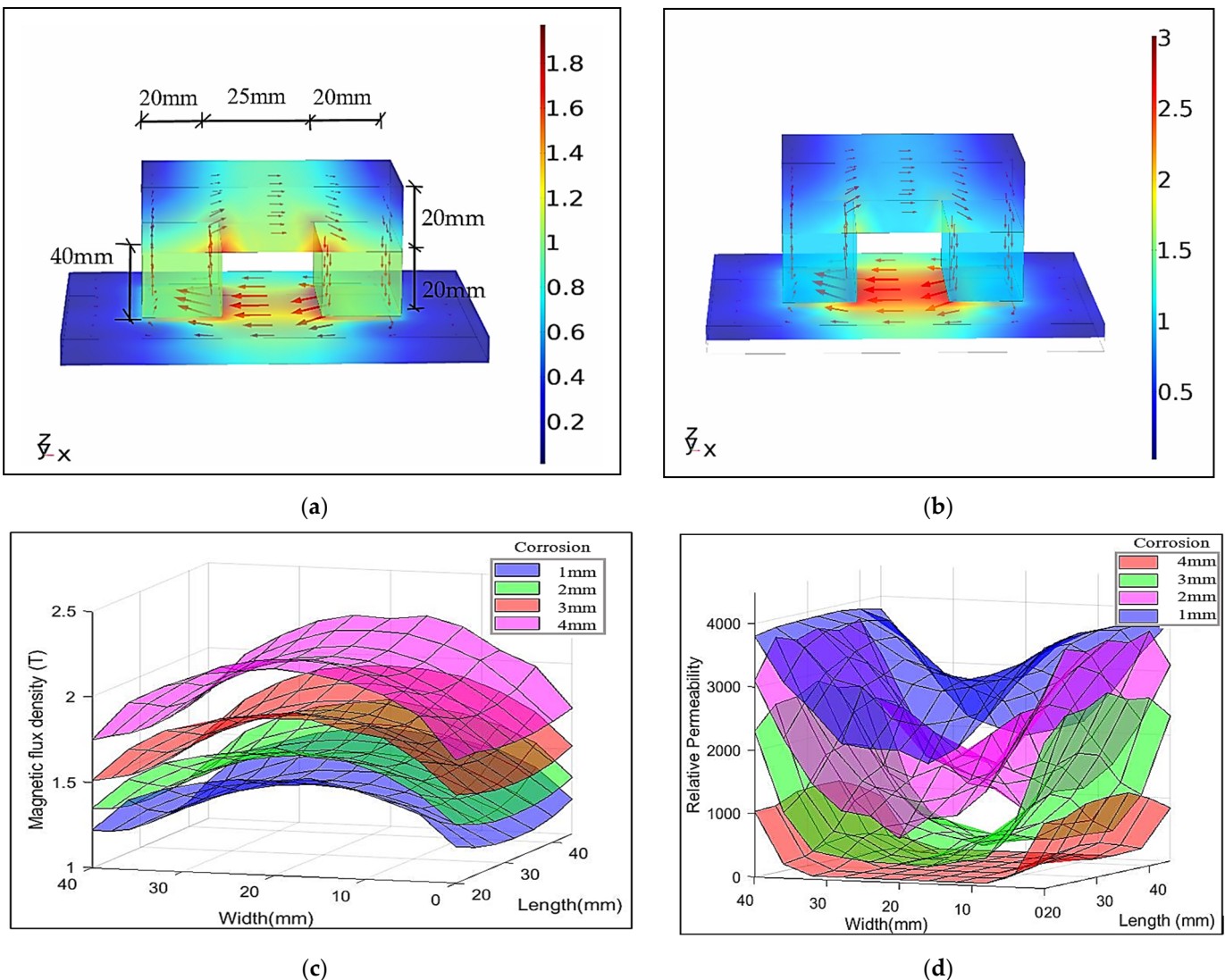

**Figure 4.** The FE model for magneto-static study. (**a**) The flux density in an 8 mm thick sample, (**b**) the increase in the flux density in a 4 mm thick plate, (**c**) the results for the flux densities in plates with different thicknesses, and (**d**) the permeability evaluated for the uniform corrosion of 1 mm–4 mm for 8 mm thick plate.

### 3.1.2. AC Modelling

The AC module of the simulation software was used for carrying out sensitivity studies. The size and number of turns in the coil are the controlled variables due to the restricted yoke size. An increase in coil size will increase the distance between the magnets, subsequently decreasing the magnetization level in the test sample. Therefore, a parametric study on the excitation frequency of the coil was conducted to optimize the sensitivity.

The Maxwell–Ampere law is used for solving the time-harmonic electromagnetic fields and is presented as:

$$\nabla \times H = (J + \frac{\partial D}{\partial t}) \qquad (10)$$

where the left side of the equation (curl of *H*) is the magnetic field produced by the stationary currents (*J*) and the displacement current ($\partial D / \partial t$) due to the deformation of magnetic vortices.

A coil, with the parameters presented in Table 3, was modeled, and the excitation frequency of the coil was swept from 1 kHz to 300 kHz, with an increment of 10 kHz.

**Table 3.** The coil parameters used for the computation of inductive reactance.

| Coil Parameters | Values |
|---|---|
| Internal diameter (ID) | 10 mm |
| Outer diameter (OD) | 18 mm |
| Height | 15 mm |
| Number of turns | 350 |

In eddy current coil simulations, the regions where eddy currents form require a very dense meh due to the skin effect. Therefore, a very refined mesh is required in the boundaries between the coil and the steel plate perpendicular to the eddy currents.

In order to simplify the problem, a 2D axisymmetric coil model was built, as shown in Figure 5.

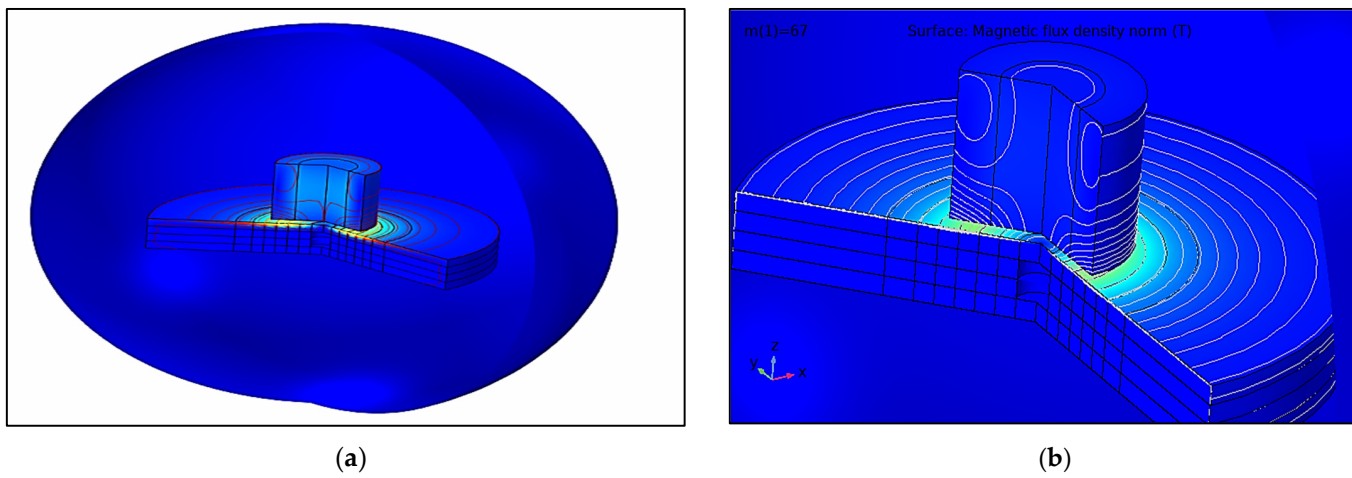

(**a**)             (**b**)

**Figure 5.** (**a**) The 2-D axisymmetric FE model, and (**b**) the solution for coil parameters at 100 kHz frequency.

The boundaries near the eddy current coil generation sites were simulated by using an adaptive mesh. It was found that the results did not converge even when the element size was as small as 0.1 mm. However, increasing the resolution of boundary regions perpendicular to eddy currents resulted in a convergence shown in Figure 6. The mesh parameters are listed in Table 4.

The permeability of the sample acquired from the first step was assigned to the elements in the model to simulate the plates of various thicknesses. Because of the wide range of variation in reactance at different frequencies, the reactance values computed for different plate thicknesses were normalized by using the reactance of 8 mm thick plate as reference value. The normalized inductive reactance values computed for all the excitation frequencies are illustrated in Figure 7.

**Table 4.** The mesh parameters obtained from mesh convergence study.

| Element Size Parameters | Values |
|---|---|
| Maximum Element Size | 0.5 mm |
| Minimum Element Size | 0.02 |
| Maximum Element growth rate | 1.1 |

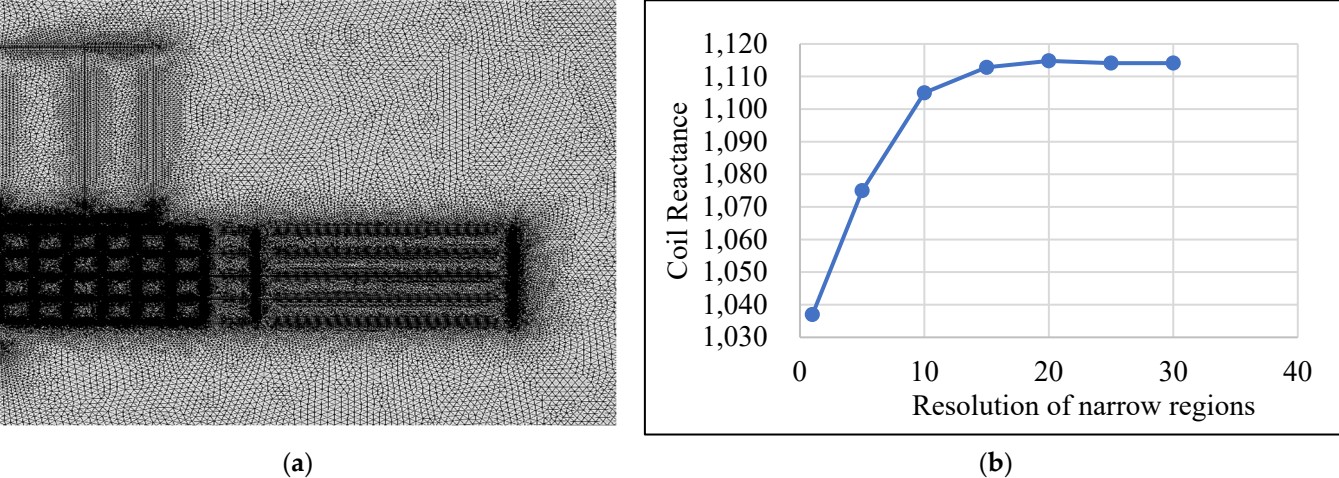

(**a**)                                                    (**b**)

**Figure 6.** (**a**) The adaptive mesh used for modeling the coil, and (**b**) the convergence of coil reactance with the increasing resolution of narrow regions.

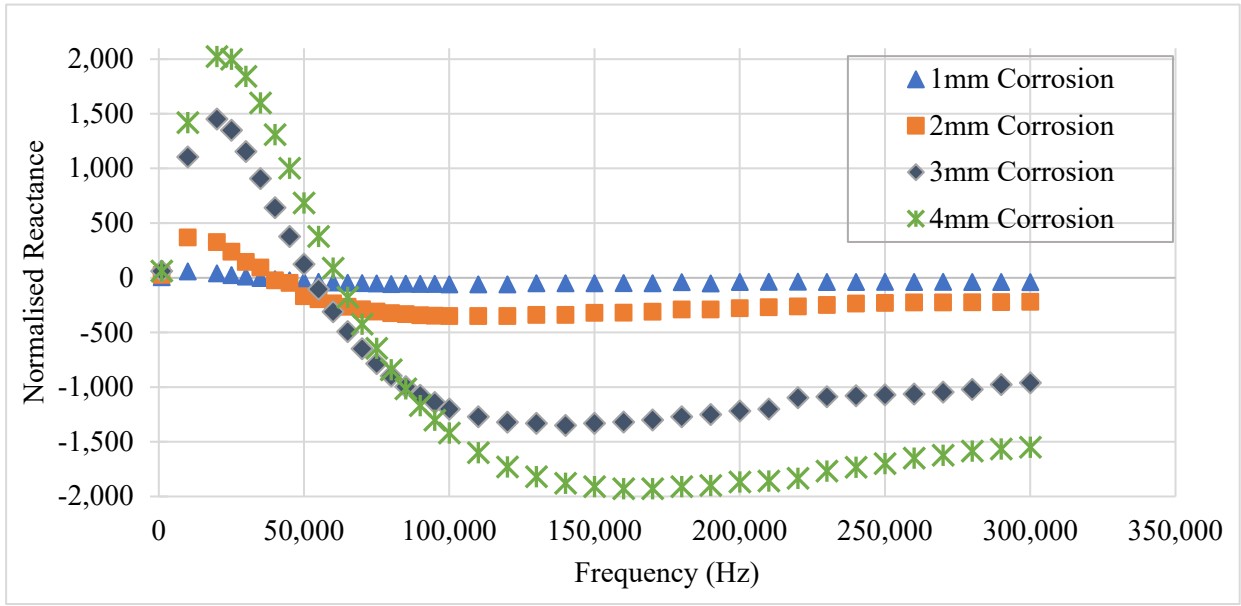

**Figure 7.** The normalized reactance values for steel plates at different excitation frequencies.

The FE model results indicate that the relationship between the sensitivity of the sensor and the excitation frequency is not monotonic. The sensitivity increases at lower frequencies initially, but it starts to decrease and becomes zero at a certain frequency called zero-crossing frequency [31,32]. It becomes increasingly negative at higher frequencies and then starts to decrease slightly. The choice of the excitation frequency is, therefore, a significant part of the design of MEC sensors.

### 3.1.3. Experimental Studies for Frequency Optimization

To validate the results obtained from the FE modeling, an experimental study was carried out on mild steel S275JR plates. An AD 5933 evaluation board was used for complex impedance measurements, which uses the commercially available high precision impedance analyzer chip [33]. The AD 5933 has a built-in 16 MHz clock and a sampling rate of 1.04 MHz It has an onboard digital signal processing (DSP) engine to carry out a discreet Fourier transform (DFT). The DFT algorithm returns a real and imaginary value for the impedance calculation. The board can measure the impedance from 100 $\Omega$ to 1 M$\Omega$

and carry out a frequency sweep from 1 kHz to 100 kHz. A higher frequency can also be measured with a small error.

The measurement procedure requires a simple inverting amplifier (resistor of known impedance) for calibrating the system. The inverting amplifier ($R_{fB}/Z_{unknown}$) senses the current in the unknown impedance and converts it into voltage [34,35]. The *Gain factor* is calculated by the formula:

$$Gain\ factor = \frac{\left\lfloor \frac{1}{Impedance} \right\rfloor}{Magnitude} \tag{11}$$

When the device under test (DUT) is connected to the board, the measurements are multiplied by the gain factor evaluated in the calibration step and then impedance is calculated. Both imaginary and real parts of impedance, along with phase, are returned.

The sensor was tested on plates with thicknesses of 4, 5, 6, 7, and 8 mm. The normalized reactance was calculated by subtracting the reactance of all the plates from the 8 mm thick plate sample. The frequency was swept from 1 kHz through to 300 kHz, with an increment of 1 kHz. The results are shown in Figure 8.

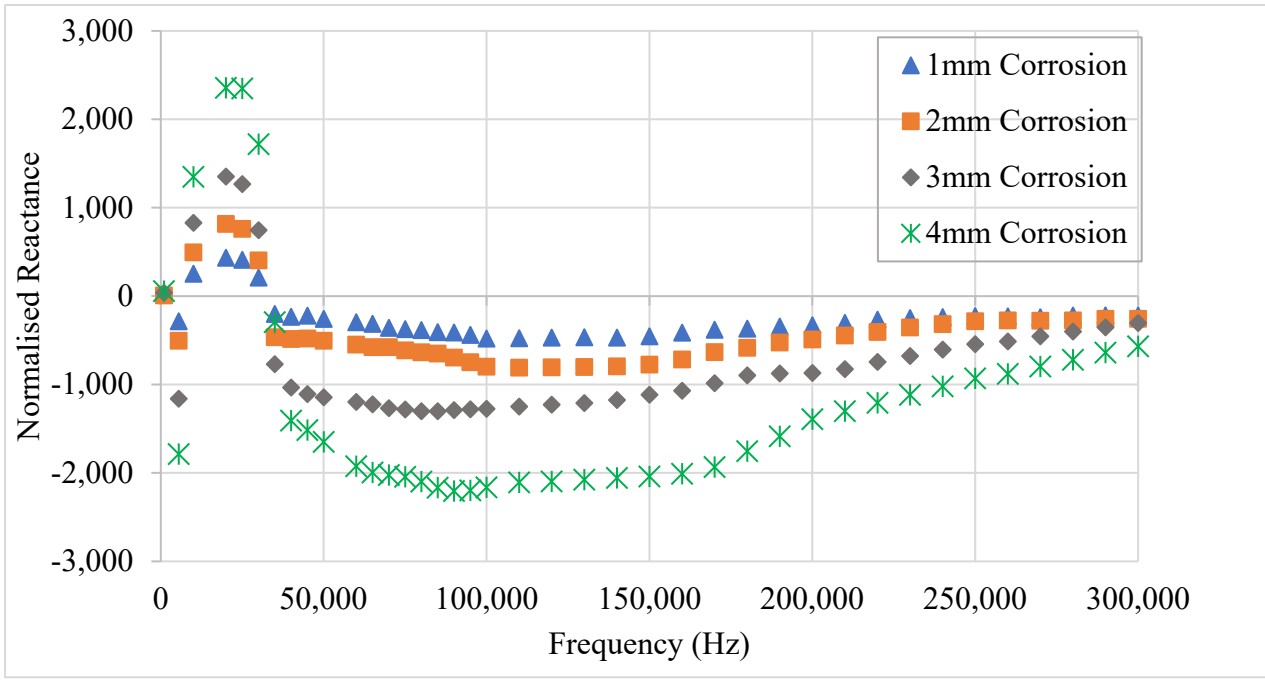

**Figure 8.** The normalized reactance for steel plates with different thicknesses obtained from experiment.

Figure 8 shows that the reactance signals follow a similar trend as those in the FE model. A theoretical explanation of the behavior of the reactance with the frequency variation when a coil is placed above the ferromagnetic sample is given below.

As evident from Figure 9, there are two sets of magnetic fields generated when an eddy current coil is placed near the ferromagnetic sample: the primary magnetic field due to the excitation current and the secondary magnetic field produced by the induced eddy currents. The primary magnetic field intensity increases with the frequency and is magnified by the permeability of the material. Meanwhile, the secondary magnetic field also increases with the increasing induced eddy currents' intensity at higher frequencies. The secondary magnetic field opposes the primary magnetic field, thus decreasing the reactance. The effect of the secondary magnetic field is not very high at lower frequencies due to the low eddy current intensity. At a certain frequency, they cancel each other out, making the reactance zero, which is called the zero-crossing frequency. Beyond this frequency, the secondary magnetic field dominates the primary magnetic field, making the reactance

appear negative. Additionally, as the saturation of eddy currents takes place at higher frequencies, the rate of change of reactance becomes constant. Both primary and secondary magnetic field strengths seem to depend on the permeability of the sample under test.

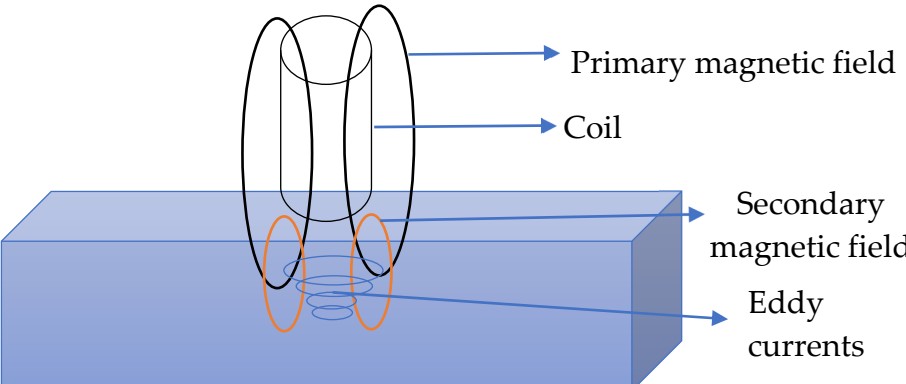

**Figure 9.** The schematic representation of the MEC principle.

From the above discussion, it is clear that there are two optimal frequencies in the sensitivity curve, and the choice of frequency depends on the type of defects and magnetic properties of the material under test. Though the FE model results agree with the experimental results at some points, there are a few dissimilarities: (1) the zero-crossing frequencies are shifted from 35 kHz to 54 kHz when the plate thickness is changed from 8 mm to 4 mm in the model. However, the same frequency shift was not observed in the experimental results; (2) 100 kHz was found to be the optimal frequency for uniform corrosion of 1 mm in 8 mm and 7 mm thick plates, but then the optimum frequency moved to 140 kHz in the simulation. In experimental studies, 100 kHz was the sensitive frequency for all plate thicknesses.

This can be explained by considering the limitations of the FE modeling of magnetic materials especially close to magnetic saturation. The FE modeling requires the assignment of complete magnetic properties of the material on point-to-point basis [36]. It is not always possible to address the complete magnetic characteristics of a material. Such detailed information on the magnetic properties of the material is not always available. The FE models are indicative rather than definitive. It was observed that the error in the FE models is more obvious near saturation because of the limited capability of the simulation software to solve for the magnetic saturation phenomena.

The results for the uniform corrosion of 1 mm in 8 mm and 7 mm plates are comparable for both the FE model and the experiments as depicted in Figure 10. Based on these results, 100 kHz was used to design the wireless MEC sensor as it has high sensitivity and is not too close to the zero-crossing point.

Though a similar trend is observed, the values of reactance computed from the FE model are different than those obtained from the tests. This can be explained by the calibration system used by AD5933 evaluation board for reactance measurement. The reference resistor can be used for the amplification of the signals at lower frequencies up to a certain value, and therefore the variation in reactance in experiments is higher than the FE results. However, careful consideration is required in the calibration process as using a very large resistor can saturate the impedance measurement circuit.

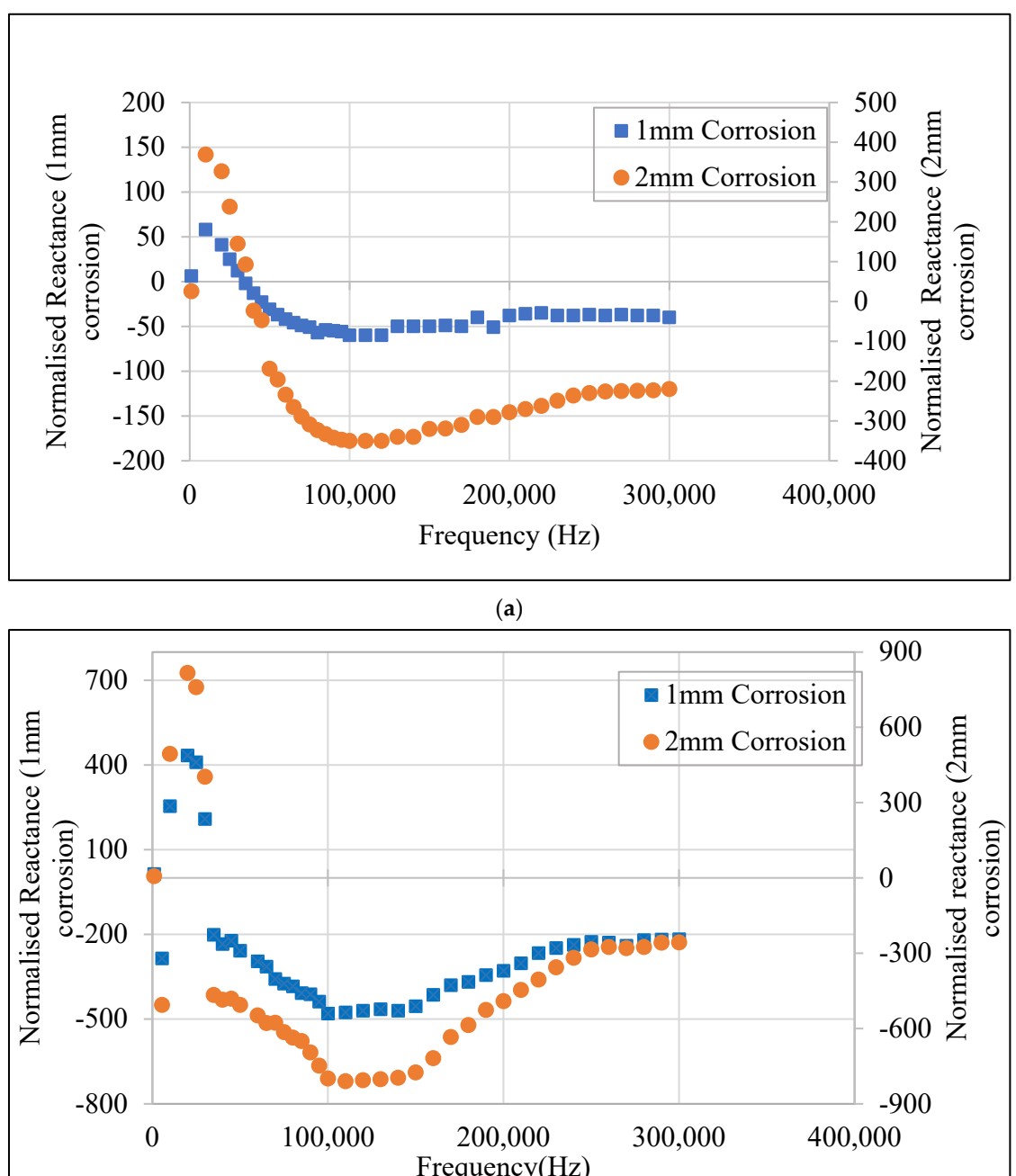

(**a**)

(**b**)

**Figure 10.** (**a**) The FE modelling normalized reactance for 1 mm and 2 mm corrosion in 8 mm thick plate. (**b**) The normalized reactance variation for 1 mm and 2 mm uniform corrosion in 8 mm thick plate obtained from experimental studies.

## 4. Design of the Wireless MEC Sensor

The AD5933 chip has an I2C interface, which enables it to connect with an external microcontroller to program the frequency sweep and store the data before transmission. The Atmega328 was used as the microcontroller. A high-pass filter was used to attenuate the frequencies lower than the operational frequency of 100 kHz. A gateway using Node MCU 3.9 and NRF24L01 transceiver was also designed to receive the signals and send them to the data cloud through Wi-Fi. The sensor was programmed to activate deep-sleep mode after sending the signals after each minute.

The AD 5933 requires a 3.3 V DC power supply and has the option to excite the DUT at four different voltage ranges. It is recommended to use the highest voltage range for less noise in the signal. Therefore, 2 *Vp-p* was used as the excitation voltage. The calibration procedure requires a known resistor to be attached between the $R_{fb}$ points, such that its value is related to the range of the unknown impedance by the following relationship.

$$R_{fb} = (Z_{min} + Z_{max})/3 \tag{12}$$

where $Z_{min}$ is the minimum unknown impedance value and $Z_{max}$ is the maximum unknown impedance value.

The block diagram of the sensor design and the prototype sensor is shown in Figure 11, Figure 12 and Figure 13, respectively.

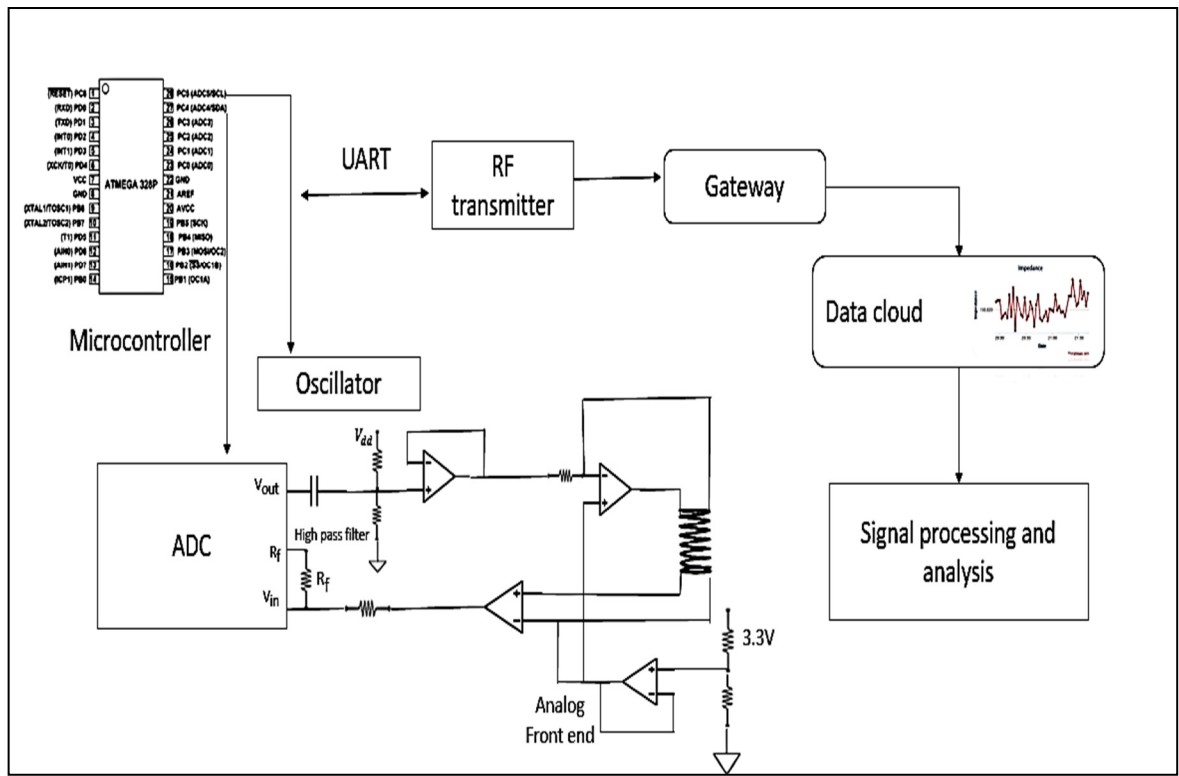

**Figure 11.** The block diagram of the design of the wireless MEC sensor.

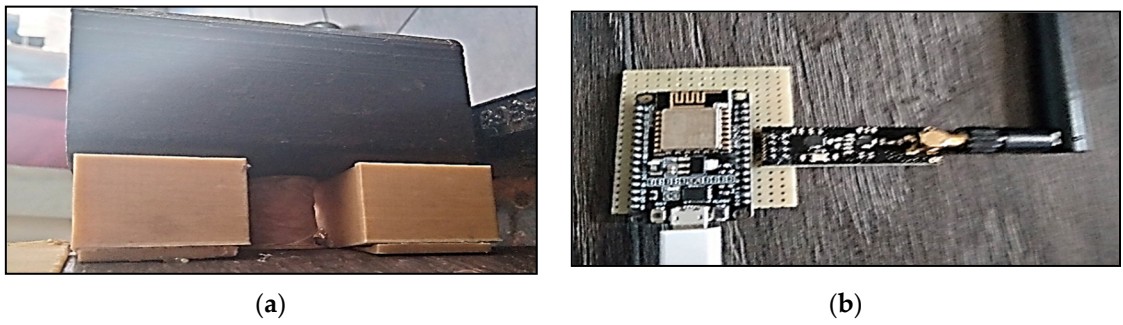

(**a**)                                    (**b**)

**Figure 12.** (**a**) The magnetization unit of the MEC sensor, and (**b**) the gateway for communication with the data cloud.

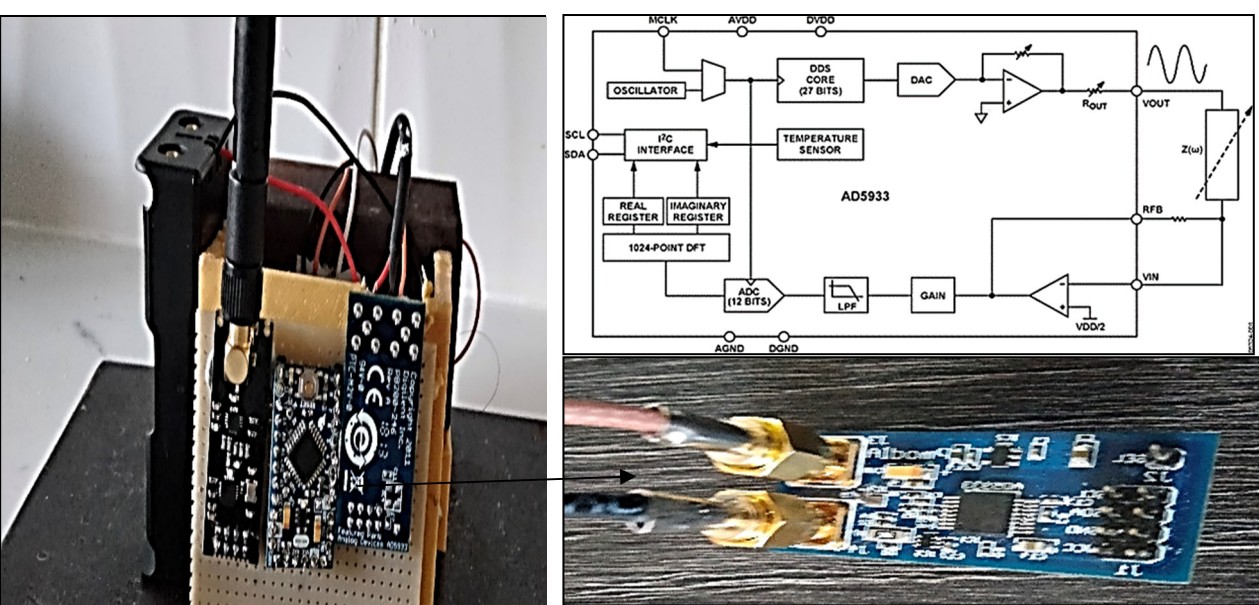

**Figure 13.** The electronic design of the sensor.

To reduce the power consumption of the sensor, after taking the measurement and signal transmission, the sensor was set to deep-sleep mode. The sensor can run for a long time on batteries as it only requires 80 μA during its operation and draws only 13 μA in deep-sleep mode

*Accelerated Corrosion Test Setup and Results*

An accelerated corrosion test was set up to test the sensor for real-time application. Diluted sulfuric acid is reported to be highly corrosive to mild steel [37]. A corrosion rate as high as 110 gm/dm$^2$/day was observed when the carbon steel samples were soaked in 10% *v/v* sulfuric acid for 24 h at room temperature [38]. The addition of water in the concentrated sulfuric acid results in an increase in the corrosion rate of carbon steel due to the solubility of ferrous sulfate protective layer formed by the acid on the steel surface [39].

Therefore, a 20% *v/v* sulfuric acid solution was used to carry out the test. An 8 mm thick S275 mild steel plate sample was kept in a cell containing the acid for 72 h. A temperature and humidity sensor (AM 2302) was also installed at the test site to record the environmental data. The sensor was programmed to record the inductive reactance after every minute and transmit it to the data cloud. The experimental setup and the corroded steel plate are shown in Figure 14.

At stages through the procedure, the corrosion products were cleaned for accelerating the corrosion process. There was a measurement noise in the signals due to the electronic components. A 10-point moving average filter was used to de-noise the signals. The results from the corrosion and AM 2302 sensors are shown in Figure 15 and Figure 16, respectively.

The graph of rate of change of reactance with respect to time in hours is shown in Figure 17. It is evident from the Figure that the corrosion rate is high at the start of the test due to the absence of corrosion products that form a protective layer around the sample.

Comparison of the data from the sensors shows that the sensor is sensitive to corrosion defects. There are no subtle differences in the signals due to the variation in temperature or humidity. There are some points where the sensor readings have a huge variation from the trend due to the movement of the sample for cleaning of the corrosion products.

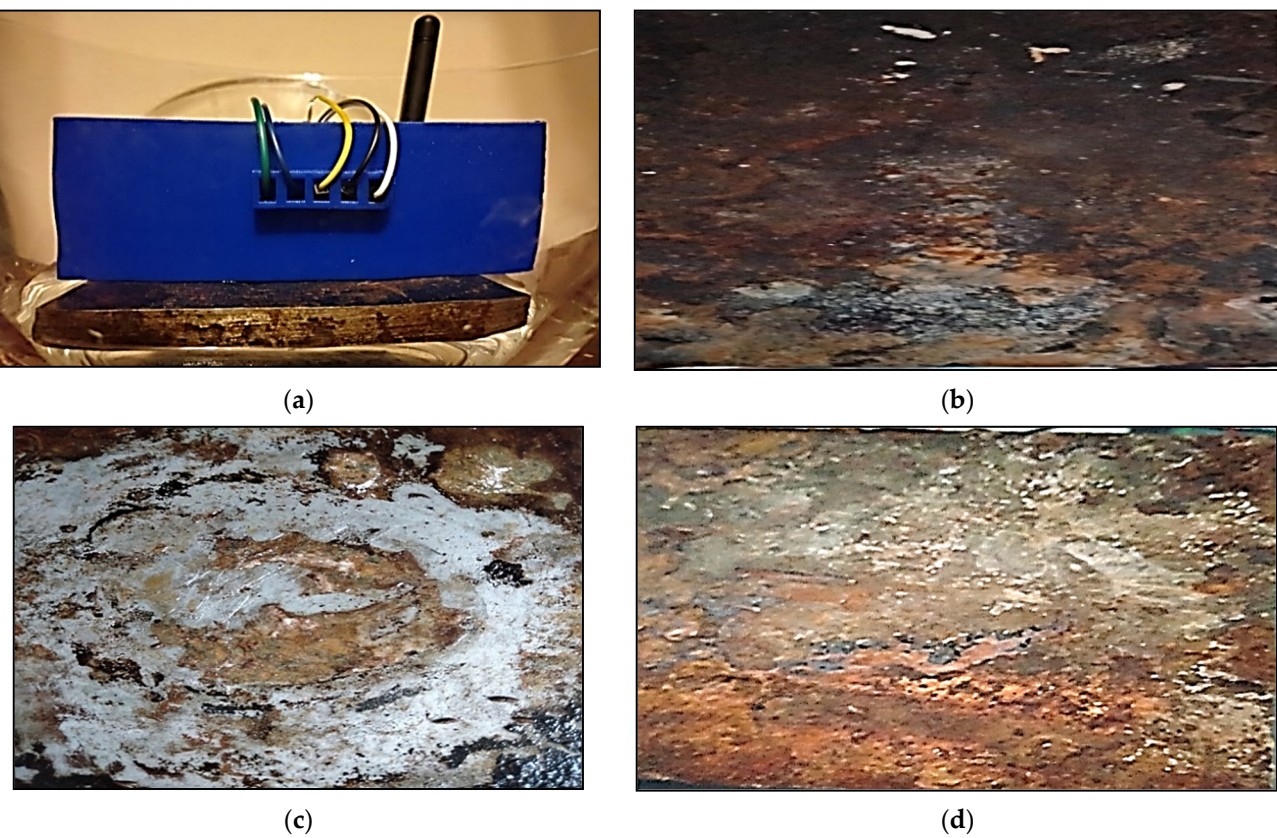

**Figure 14.** Experimental set up and corrosion in the steel plate at different stages: (**a**) experimental set up, (**b**) corrosion after 12 h, (**c**) corrosion after 24 h, and (**d**) corrosion after 72 h.

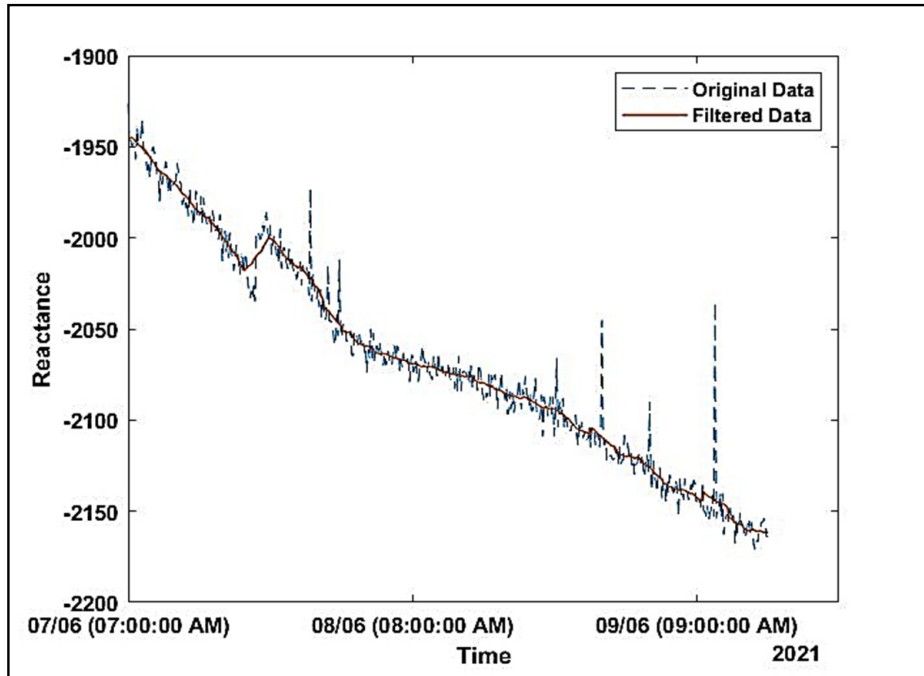

**Figure 15.** The signals of the MEC sensor recorded for 72 h.

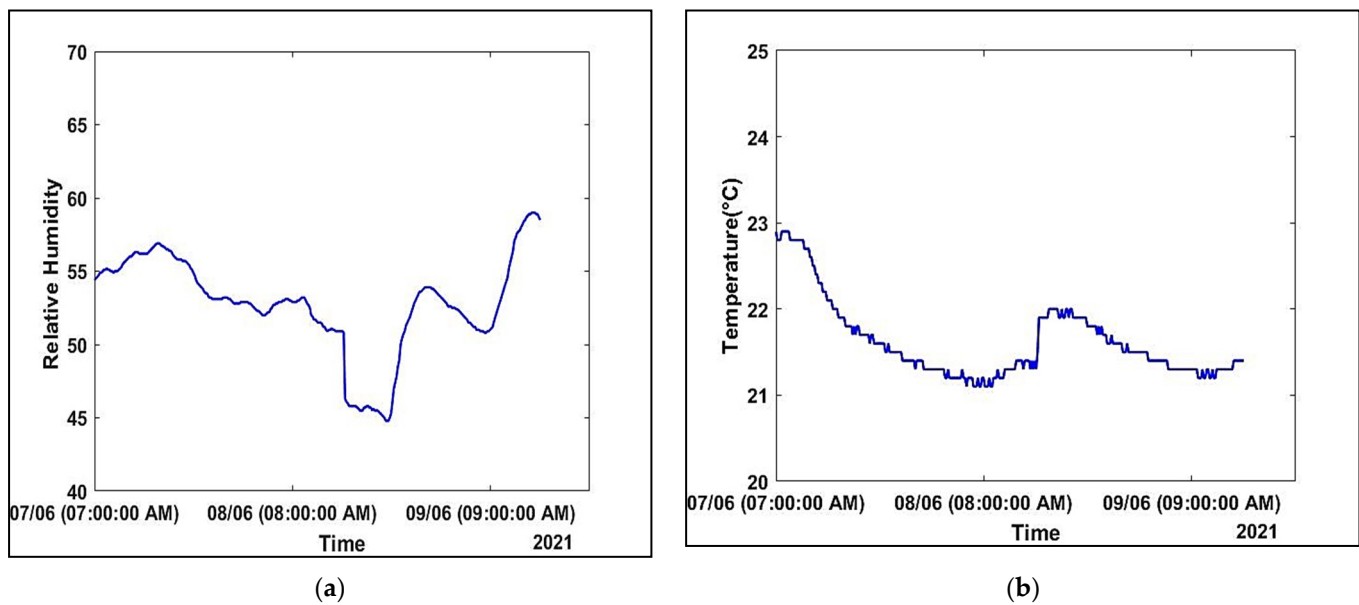

**Figure 16.** (**a**) The relative humidity (%RH) values, and (**b**) temperature (°C) recorded throughout the procedure.

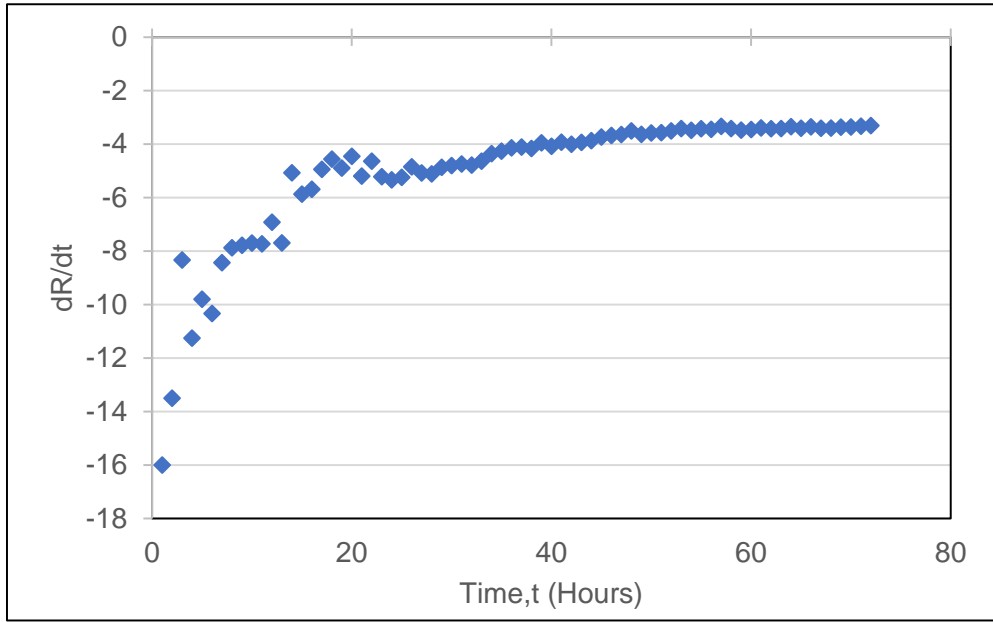

**Figure 17.** The graph showing the rate of change of corrosion per hour throughout the test.

## 5. Discussion

The MEC technique is commercially used for the detection of buried defects such as corrosion and cracks. The efficacy of the MEC sensors in in-line and online inspection tools has been well established. The prospects of using a wireless magnetic eddy current sensor as a permanently installed sensor for continuous corrosion monitoring were tested using an accelerated corrosion test. The sensor design was optimized using finite element modeling to achieve higher sensitivity without compromising on the small size.

The MEC method relies on sensing the perturbations in the permeability of the test sample due to defects. The inductive reactance of the coil-excited sinusoid-ally is affected by the changes in permeability. It was observed that a wide range of excitation frequencies are discussed in the literature for the detection of surface and buried defects in ferromagnetic structures. Most of the research on MEC sensors is focused on the optimization

of the magnetization level and the defect characterization and quantification, by using different signal parameters. Through FE simulations and experiments, it was found that the sensor's excitation frequency has a major effect on its detection capabilities. The finite element modeling revealed that the relationship between sensitivity and frequency is not monotonous. The reactance values increase positively at lower frequencies and decrease considerably with the increasing frequency. At a certain frequency called zero-crossing frequency, the reactance becomes zero and then starts to increase negatively.

The theoretical studies show that the reactance of the coil is dependent on two sets of magnetic fields set up by the coil. The primary magnetic field is due to the excitation current and secondary magnetic fields because of the induced eddy currents. Both of the fields are magnified by the high permeability of the ferromagnetic sample and are affected by the variation in permeability. They oppose each other due to the opposite nature of the currents producing the fields. At lower frequencies, the primary magnetic field dominates the other and vice versa. The study indicates that two optimal frequencies can be used for inspection and the frequencies near the zero-crossing range need to be avoided.

Furthermore, the FE model and experimental results were found to disagree at some points. The zero-crossing frequencies and the optimal frequencies for FE models and test results were not the same. The results had some similarities for thick plates with less magnetization level, but the error increased when the magnetic saturation was achieved in the plates due to increased corrosion. This can be explained by the limited capability of the simulation software for modeling magnetic saturation phenomena.

The excitation frequency of 100 kHz was used for the design of the wireless sensor due to the following reasons: (1) the sensitivity was maximum at 100 kHz for both the FE model and experimental results for thick plates, and (2) the sensitivity near the 100 kHz range did not experience significant fluctuations.

The wireless sensor designed by interfacing AD 5933 with a microcontroller unit was tested by conducting accelerated corrosion on a mild steel plate. Along with sensor signals, the environmental data was also recorded. The comparison of the signals revealed that the sensor was sensitive to corrosion and the signals were not affected by temperature or humidity level changes.

## 6. Conclusions

A novel wireless magnetic eddy current sensor was developed and tested using an accelerated corrosion test on mild steel. Sensitivity studies were carried out using FE simulation to optimize the size of the probe. The results of the FE models were validated by measuring the reactance of the sensor for different plate thicknesses for a range of excitation frequencies. The comparison of FE simulation and experimental investigation results showed that the excitation frequency is a significant parameter in the sensor design. The relationship between the excitation frequency and sensitivity of the sensor was discussed using the eddy current principle.

Based on the findings from the FE simulation and tests, the wireless sensor was designed and tested in an accelerated corrosion setup. The results of the study indicated that the corrosion sensor can potentially be used as a permanent corrosion monitoring sensor.

## 7. Future Work

The focus of this study was on the development of an MEC sensor for permanent installation for detection of internal corrosion. The study can further be extended to quantify the corrosion defects using the different signal parameters. The development of multi frequency sensor can be considered in the future as the frequency is highly related to the permeability of the sample under test and may help in characterizing the nature of defects. The effect of frequency on the sensitivity of the sensor is critical for defect detection. This can further be studied in the future to correlate the optimal frequency with the material magnetic properties and nature of the defects. The dissimilarities found between the FE

model and the experimental results can further be investigated to decrease the error by introducing the different approaches for simulating the magnetic saturation modeling.

**Author Contributions:** Conceptualization, R.W., J.R and M.O.T.; methodology, R.W.; software, R.W.; validation, R.W.; formal analysis, R.W.; investigation, R.W.; resources, J.R. and R.M.; data curation, R.W.; writing—original draft preparation, R.W.; writing—review and editing, R.W.; visualization, R.W.; supervision. M.O.T., J.R., G.S. and R.M.; project administration, R.M.; funding acquisition, R.M. All authors have read and agreed to the published version of the manuscript.

**Funding:** This research was funded by Lloyds Register Foundation and TWI LTD, CB21 6AL, Cambridge, UK.

**Institutional Review Board Statement:** Not applicable.

**Informed Consent Statement:** Not applicable.

**Data Availability Statement:** Available on request due to privacy and ethical issues.

**Acknowledgments:** This research was made possible by the sponsorship and support of Lloyds Register Foundation. The work was enabled through, and undertaken at, the National Structural Integrity Research Centre (NSIRC), a postgraduate engineering facility for industry-led research into structural integrity established and managed by TWI through a network of both national and international Universities.

**Conflicts of Interest:** The authors declare no conflict of interest.

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
