# Peer review of "Development of Permanently Installed Magnetic Eddy Current Sensor for Corrosion Monitoring of Ferromagnetic Pipelines"

_applsci, doi:10.3390/app12031037_

Round 1
Reviewer 1 Report
Dear authors,
I found the paper well designed and interesting to read.
On the other hand there are many misprints...
Overall the paper it is not clear how may types of sample thicknesses you tested...
It seems five : 8 --> 4 but figure 2 report only four...
However figure 2d) caption is wrong...
Figure 5 should be 4 and so on for the others...Please add the grid with ticks
line 222 INCREASE slightly
Figure 6 Please use same markers for same samples...
Figure 7-8 A discussion should be added in Reactance value: Experiment is SEVEN TIMES simulation...not so equal behaviour as it is stated
Then, please add the graph of 8 mm sample that is neglected in all the paper figures...
Figure 13: Please add the derivative of the Reactance, it seems to increase again after a couple of days...Very strange according to standard corrosion models...Please add more details of the acid sufuric bath experimental methods
Finally Discussion Section is Already Conclusion
and Conclusion section should be deleted. Future work is OK
Reviewer 2 Report
The paper presents interesting results concerning the optimization of the wireless MEC system. The theoretical background and results are generally well described. However, some information on the parameters of the simulated system is missing. The simulation part is the weakest part of the work and may need to be improved or at least described in more detail. I enclose my detailed comments below:
l. 39 – the acronym NDE (non-destructive evaluation as I suppose) should be explained here
l. 119 – “Magnetization and relative permeabilityas a function of applied field”. – please correct the typo
l. 121-125 – “When a steel sample is magnetized beyond the point (a) in the Figure (1), there is a leakage magnetic field ∅???, whereas the magnetic field inside the un-corroded sample is , ∅1. If there is a decrease in the wall thickness of the sample owing to corrosion, there will be an increase in the ∅??? and the flux density inside the sample will rise from ∅1 to ∅2 correspondingly decreasing the magnetic permeability from ?1 ?? ?2.” – Reasoning presented by the authors is generally correct, however, I propose them to support it with a comparison of reluctance equations for those two ‘magnetic circuits’. Moreover, using the symbol of ‘∅’ can mislead the reader familiar with magnetism, because it is associated with the magnetic flux, which is the product of B*S and as such cannot be marked on the horizontal (H – magnetic field intensity) axis of the graph.
l. 133 – should be “mild steel”
l. 135-137 – “Secondly, the eddy currents field, set up in the sample due to the time-harmonic magnetic field of the coil, will be affected by the perturbations in the permeability caused by the corrosion defects”. – I guess what the authors meant, but they should be more precise. Said change of the permeability is caused by the decrease in the reluctance, which in turn results from the decrease in the cross-section area of the wall.
l. 172 – equation (4) does not represent the curl of the magnetic scalar potential as declared in the text
l. 179-180 – “It was found that a minimum flux density of 1.4 T in the sample can give a good sensitivity using a probe size of 180 40x65x40mm”. – As a reader I think it would be worth to add a figure defining geometry of your probe and symbolically its dimensions. Moreover, it is not clear what the parameters were the variables during optimization and what the objective functions were. Readers also do not know what magnets were simulated and what were magnetic properties of the magnetic bridge.
l. 201-203 – “In electromagnetic wave propagation studies, it is critical that the mesh size of the model is less than the wavelength of the wave to ensure convergence of results. At higher frequencies this can be a problem due to a very small element size requirement.” – It is not the case described in this study. For the highest frequency of 300 kHz the associated wavelength equals 1000 m (!). Mesh refinement, however, should be applied especially on the outer surfaces of conductive objects due to the skin effect.
l. 204 – “Therefore, a simplified 2D axisymmetric coil model was built as shown in Figure 4”. – The question then arises why the authors didn't simplify their analysis to axisymmetric geometry. There is no benefits from performing a 3D study if you have perfect axisymmetric geometry.
l. 224 – Section 3.1.3 - A reader should be also informed what were parameters of the coil used in the experiment.
l. 244-245 – “The frequency was swept from 1kHz through to 300kHz with an increment of 1kHz” – Results presented in Figure 6 show that the increment was rather variable and it was dependent on a frequency range.
l. 262-275 - I believe that this theoretical explanation could be enriched if the authors also took into account the frequency-changing relationship between the depth of penetration and the thickness of the sample. My experience shows that the extreme negative reactance can occur when the penetration depth becomes comparable to the thickness of the sample.
l. 286-293 – Observed discrepancies can be also associated with the mesh density, so it is worth to use an adaptive meshing and/or check sensitivity of the simulation results to the mesh refinement. Another source of the discrepancies is usually difference between real properties of the setup components and those used in the simulation. So if the authors used built-in material libraries in their simulation, it could be also a source of the discrepancies.
l. 324 – Caption of Figure 10 - please correct the typo
Round 2
Reviewer 2 Report
The authors responded to the reviewer's doubts in a satisfactory manner and responded to his comments. I have no additional comments.